# Evaluation of Lipopolysaccharide and Interleukin-6 as Useful Screening Tool for Chronic Endometritis

**DOI:** 10.3390/ijms25042017

**Published:** 2024-02-07

**Authors:** Erina Yoneda, Sangwoo Kim, Kisaki Tomita, Takashi Minase, Mitsunori Kayano, Hiroyuki Watanabe, Masafumi Tetsuka, Motoki Sasaki, Hiroshi Iwayama, Hideomi Sanai, Yuki Muranishi

**Affiliations:** 1Department of Life and Food Science, Obihiro University of Agriculture and Veterinary Medicine, Obihiro 080-8555, Hokkaido, Japans21180002@st.obihiro.ac.jp (S.K.); s23170036@st.obihiro.ac.jp (K.T.); hiwatanabe@obihiro.ac.jp (H.W.); mtetsuka@obihiro.ac.jp (M.T.); 2Sapporo Clinical Laboratory Inc., Sapporo 060-0005, Hokkaido, Japan; minase-t@rf7.so-net.ne.jp; 3Research Center for Global Agro-Medicine, Obihiro University of Agriculture and Veterinary Medicine, Obihiro 080-8555, Hokkaido, Japan; kayano@obihiro.ac.jp; 4Department of Veterinary Medicine, Obihiro University of Agriculture and Veterinary Medicine, Obihiro 080-8555, Hokkaido, Japan; sasakim@obihiro.ac.jp; 5Obihiro ART Clinic, Obihiro 080-0803, Hokkaido, Japan; keiaiivf@keiai.or.jp (H.I.); hsanai@keiai.or.jp (H.S.); 6Laboratory for Molecular and Developmental Biology, Institute for Protein Research, Suita 565-0871, Osaka, Japan

**Keywords:** CD138, chronic endometritis, IL-6, inflammation, lipopolysaccharide

## Abstract

Universal diagnostic criteria for chronic endometritis (CE) have not been established due to differences in study design among researchers and a lack of typical clinical cases. Lipopolysaccharides (LPSs) have been reported to cause inflammation in the reproductive systems of several animals. This study aimed to elucidate the influence of LPS in the pathogenesis of CE in humans. We investigated whether LPS affected cytokine production and cell proliferation in the endometrium using in vivo and in vitro experiments. LPS concentrations were analyzed between control and CE patients using endometrial tissues. LPS administration stimulated the proliferation of EM-E6/E7 cells derived from human endometrial cells. High LPS concentrations were detected in CE patients. LPS concentration was found to correlate with IL-6 gene expression in the endometrium. Inflammation signaling evoked by LPS led to the onset of CE, since LPS stimulates inflammatory responses and cell cycles in the endometrium. We identified LPS and IL-6 as suitable candidate markers for the diagnosis of CE.

## 1. Introduction

Chronic endometritis (CE) is one of the causes of unexplained infertility and repeated implantation failure [1,2]. CE is an inflammatory disease of the endometrium, which is characterized by mucosal edema, polyps, and abnormal plasma cell infiltration [3]. A retrospective cohort study of 1551 premenopausal women reported a 24.4% incidence of CE [4]; however, precise diagnostic criteria for CE have not yet been established. Chronic inflammation of the endometrium may be accompanied by symptoms such as pelvic pain, irregular genital bleeding, and intercourse pain; however, it is often asymptomatic and difficult to diagnose [5,6].

In general, CE is diagnosed using hysteroscopy and pathologic examination. Currently, next-generation sequencing (NGS) analysis is focused on the bacterial flora of the vagina and uterus [5,7]. Hysteroscopy provides subjective information by the physician, which may not confirm the clinical findings. The pathological diagnosis of CE involves staining for plasma cells in endometrial tissue, which is frequently performed using CD138 immunostaining. However, the histological method of CD138 cannot be used in all the scenarios due to lack of consensus on a threshold for the definition of CE [8]. Additionally, the efficiency of CD138 detection depends on the timing of the menstrual cycle, which influences endometrial proliferation [5,9]. Therefore, both hysteroscopy and pathological examination remain uncertain for the diagnosis of CE.

Gram-negative bacteria have an endotoxin (lipopolysaccharide: LPS) in the outer membrane of the cell wall [10]. LPS has been reported to cause inflammation in the reproductive systems of several animals [11]. Intravaginal administration of LPS was used to create a model of acute endometritis in mice [12]. LPS administration causes extensive inflammatory macrophage infiltration of uterine tissue, and it also increases the expression of *Il-1β*, *Tnfα* and other genes in mice [13]. In humans, LPS affects the trophoblastic spheroids and endometrial epithelial cells and decreases uterine receptivity [14]. On the other hand, few human studies have reported that LPS is related to endometrial inflammation, which has been associated with infertility and repeated implantation failure [1]. Furthermore, another study has shown the effect of proinflammatory cytokines (IL-6, IL-1β and TNFα) on menstrual effluent CE patients. The levels of these cytokines were significantly higher in patients with CE compared to controls [15]. An investigation of cytokine levels can be used as a non-invasive method for the detection of CE. However, the pathogenic mechanism of CE remains unknown, and there is no consensus on its diagnostic criteria. In this study, the effects of LPS on the inflammatory response and reproductive function of the human endometrium were investigated to elucidate the mechanisms of CE using endometrial tissues. In addition, we aimed to discern potential markers for CE and mechanisms involved in the molecular functioning of the endometrium.

## 2. Results

### 2.1. Effect of LPS on Proliferation of EM Cells

We performed an experiment to estimate the effect of LPS, which is involved in inflammation and cell cycle in vitro using EM-E6/E7-hTERT-2 cells (EM cells) that originate from endometrial gland epithelial cells [16]. *IL-1β* is a marker of inflammation with increased expression at 4 h by LPS administration in EM cells (Appendix A). Gene expression analysis was performed for inflammatory (*IL-1β*, *IL-6*, *CD14*, *TLR4*, *CD138*) and cell cycle markers (*CyclinD1*, *p27*, *p53*, *Ki67*) 4 h after LPS administration (Figure 1). The expression of the *IL-1β* gene was increased 5.5-fold at 10^3^ ng/mL and 22-fold at 10^4^ ng/mL LPS concentrations compared to the controls (*p* < 0.05, Figure 1A). The expression of the *IL-6* gene was 69-fold higher compared to the controls at 10^4^ ng/mL LPS concentrations (*p* < 0.05). The expression of *TLR4* was significantly increased at concentrations of 10 ng/mL and 10^2^ ng/mL LPS (*p* < 0.05, Figure 1B). EM cells were examined for gene expression related to cell cycle and proliferation. The expression of the *p27* and *Ki67* genes was significantly increased at concentrations of 10, 10^2^ and 10^3^ ng/mL LPS (*p* < 0.05, Figure 1C). LPS did not alter the expression of *CD14*, *CD138*, *CyclinD1* and *p53.*

### 2.2. LPS Promotes Cell Proliferation of EM Cells

LPS increased the expression of *Ki67* in Figure 1, which led to the hypothesis that the proliferation of EM cells was induced by LPS. CCK8 assay indicated that the viability of EM cells increased after LPS administration at a concentration of 10 ng/mL LPS (*p* < 0.05, Figure 2).

### 2.3. Effects of LPS Administration on Mouse Uterus In Vivo

Furthermore, to examine whether LPS affects the mechanism of inflammation and proliferation in vivo, we performed an experiment using a mouse model of the inflammatory environment of the uterus. The chronic inflammatory condition in mice was consecutively induced by administering LPS (1 mg/mL) once a week for 2 weeks. After a week, the samples were collected and analyzed for inflammation. LPS-injected mice showed no significant differences in physiology, including body weight (BW) and LPS concentrations in plasma and uterine supernatants, compared to controls (Appendix A). Additionally, no hyperemia or histological lesions were observed in the uterus post-LPS injection (Figure 3A and Appendix A). The IHC staining of CD138, which is utilized for the diagnosis of CE, demonstrated conspicuous positive signals in the endometrium of the group of mice administered with LPS in the mouse uterus (Figure 3A). The overall area, including the uterine cavity, did not differ between the control and LPS groups (*p* = 0.13). However, a trend of decrease was indicated in the functional layer area, including the functional layer within the uterine cavity, in the LPS group compared to the control group (*p* < 0.1, Figure 3B). LPS stimulation increased the expression of *Tlr4* in mouse uterus compared to the controls (*p* < 0.05, Figure 3C). The expression of *Ki67*, which is a proliferation marker, was significantly lower in the LPS group than in the control group (*p* < 0.05). LPS did not alter the expression level of *Il-6* and *CD138* (*p* = 0.25, *p* = 0.12). Although the IHC staining of CD138 observed clearly increased the signal of CD138-positive cells, there was no discernible difference in the gene expression of *CD138* in uterine tissues between the LPS group and the control group. Despite the detected positive signal in the IHC staining due to LPS stimulation in the mouse uterus, distinguishing the gene expression of *CD138* between the LPS group and the control group proved challenging. Furthermore, the group of LPS stimulation into the uterine exhibited a tendency toward reducing the functional layer in the endometrium and the decrease in the gene expression of Ki67 compared to the control group. These findings indicate a disruption in the normalcy of the endometrium, highlighting the impact of consecutively induced inflammatory conditions in the CE mouse model.

### 2.4. Correlation between LPS and Inflammatory Genes in Human Endometrial Tissues

The gene expression levels of *TLR4*, *CD138* and *IL-6*, which are known LPS inflammatory markers, were compared between CE patients and controls (Figure 4). The expression of *TLR4* and *IL-6* was significantly higher in patients with CE than in control patients (*p* < 0.05), whereas no significant difference was found in the expression of *CD138* (*p* = 0.22).

### 2.5. Correlation between LPS and Cell Cycle Genes in Human Endometrial Tissues

The results from EM cells indicated that LPS administration increased the expression of cell cycle markers. We next assessed whether LPS was associated with cell cycle progression in human endometrial tissues. The expression of *CyclinD1* and *p27* was not significantly different (*p* = 0.48, *p* = 0.25, Figure 5). In contrast, *p53* and *Ki67* showed significantly lower gene expression in patients with CE than in controls (*p* < 0.05).

### 2.6. Relationship between LPS and CE

Figure 6 shows a representative photomicrograph of CD138 immunostaining of uterine sample from a patient with suspected CE (Figure 6A). LPS concentrations in the endometrium were significantly higher in the patients with CE than in controls (*p* < 0.05, Figure 6B,C). No significant differences were observed between the two groups in context of the age and smoking status (Appendix A). To further investigate the effect of LPS-induced inflammation, known inflammatory markers were examined for their correlation with LPS concentrations in the endometrium (Appendix A) The correlation coefficient between LPS and the gene expression of *IL-6* was indicated as rs = 0.62, stipulating a moderate positive correlation. The expression of *Ki67* indicated a negative correlation with LPS concentration in endometrial tissues (rs = −0.44, Appendix A).

## 3. Discussion

Our study indicated that high LPS concentrations were detected in endometrial tissues of patients with suspected CE. Among various inflammatory markers, we verified that IL-6 expression correlated with LPS, which suggested that LPS increased inflammatory responses in the endometrium. Our findings raise the possibility that LPS and IL-6 are potential diagnostic criteria for the diagnosis of CE. We observed that inflammation promoted the proliferation of EM cells in vitro, whereas a reduction in the endometrium was observed between humans and mice. Here, we provide evidence that LPS induces inflammation and cell cycle disruption in the endometrium and that LPS may continuously lead to the symptoms of CE.

Recent studies have reported the likelihood that vaginal and uterine bacteria can cause endometritis [17,18], suggesting that the LPS of bacteria such as *E. coli* may be responsible for causing CE. Although LPS has been detected in the menstrual effluent of sterile women, as well as in menstrual blood and peritoneal fluid from women with endometriosis [19,20], the existence of LPS in the endometrium has not been identified in gynecological research. High concentrations of LPS were detected in the endometrium of patients with CE in this experiment, and the results suggested that LPS increased the production of cytokines, including IL-6, and initiated inflammation in the endometrium.

Our findings were consistent with those of other studies, which showed that the expression of proinflammatory cytokines, such as IL-6 and IL-1β, was significantly higher in menstrual effluent patients with CE [14]. IL-6 is a well-known inflammatory marker of LPS [10,21], and IL-6 has also been shown to be a useful diagnostic marker for endometriosis, which is a chronic inflammatory disease [22]. In bovine endometrial cells, the upregulation of IL-6 and IL-8 due to LPS administration is considered a characteristic feature of inflammation [23]. Other studies reported that a mouse model of endometritis that was injected with LPS in the uterus showed that LPS significantly increased the expression of *Il-6* [12]. However, no difference was observed in *Il-6* gene expression after 24 h of LPS administration to mouse uterus in this study. This suggests that it was too late to confirm the timing of inflammatory response after LPS administration [24]. Our study suggests that continuous LPS stimulation causes inflammation in human endometrial tissues as LPS correlates with IL-6 expression. Similarly, the results of EM cells showed that LPS administration rapidly upregulated the expression of *IL-6* after 4 h. Hence, our study provides evidence that cytokine IL-6 was activated by LPS in the endometrium to facilitate the symptoms of CE.

Previous studies have reported that IL-6 stimulates LPS signal transduction through CD14 and TLR4 [25,26]; however, CD14 did not correlate with LPS levels in human endometrial tissue in this study. Other studies have reported that CD14 does not necessarily interact directly with TLR4 [27,28], and it was suggested that inflammatory cascades other than CD14 may be present. We revealed that LPS caused increased *IL-6* and *TLR4* gene expression in the endometrium in our experiments using human endometrial tissue, EM cells, and mice. Therefore, the results indicated that the presence of LPS might elevate cytokines and cause CE through the TLR4 cascade. A previous study reported that LPS in menstrual blood was involved in TLR4-mediated endometrial proliferation [20]. We also showed that LPS upregulated *TLR4* expression, which affected *Ki67* expression and disrupted the cell cycle in the endometrium. Our results suggested that LPS-mediated TLR4 affects cell proliferation in the endometrium. Abnormal proliferation of the endometrium can lead to gynecological problems, such as endometriosis, endometrial hyperplasia, unexplained infertility, repeated implantation failure and repeated pregnancy loss [29,30].

The diagnosis of chronic endometritis is complicated, there are mainly three detection methods, and cross-testing will also be performed according to the clinical situation The diagnostic methods for CE include the observation of uterine hyperemia by hysteroscopy, histological examination using CD138, which is the most specific indicator of plasma cells, and bacterial culture to identify the cause of infection [31,32,33,34]. In hysteroscopy, the subjective judgment of the physician affects the test results. There are various problems associated with the use of CD138 for the diagnosis of CE. Healthy women have CD138 in the endometrial stroma [35], and the expression of CD138 is dependent on the menstrual cycle [5,36]. In this study, CD138 was used as an inflammatory marker, although it was difficult to distinguish between CE patients and controls. Therefore, we considered IL-6 to be a more appropriate biomarker for differential diagnosis for CE than CD138 because CD138 is known to be involved in the pathogenesis of inflammatory disease, which leads to changes in IL-6 levels [37]. Some bacterial species may pose challenges in cultivation, and certain bacteria can influence the accuracy of culture results. These considerations should be thoroughly discussed in the context of chronic endometrial diseases during further exploration. Additionally, it is essential to incorporate information on NGS analysis as it relates to the study of CE [5]. NGS provides a powerful tool for comprehensive microbial profiling and may offer valuable insights into the microbial composition and diversity associated with CE.

We observed significant differences in estradiol and progesterone hormone concentrations in the supernatant of human endometrial tissue between CE patients and controls (Appendix A). This result was considered to be influenced by the menstrual cycle rather than by LPS. We did not observe any relationship among LPS, BMI (body mass index), bacteria testing and smoking status in this study. Samples were collected to examine the effect of LPS on clinical data.

Our results showed that LPS increased the expression of *p27* and *Ki67*, and it also increased the proliferation of EM cells. It is known that NF-κB, one of the signaling pathways activated by LPS, promotes cell proliferation in the endometrium [38]. However, we found that LPS decreased the expression of proliferation marker *Ki67* in human endometrial tissues and mice. Another experiment using cultured cells showed that LPS administration results in abnormal cell cycles [39]. Our experiments using humans and mice in vivo were considered to downregulate cell proliferation, because the endometrium was continuously exposed to LPS, and chronic inflammation in vivo may cause a decrease in endometrial cells after acute inflammation. The mouse model of chronic inflammation induced by LPS administration showed an increase in the myometrium. It is thought that abnormal cell proliferation is due to endometrial inflammation, because B cells which exist in the basal layer near the myometrium may induce plasma cells to exert the effect of LPS [40]. Although this experiment did not confirm the phenomenon of cell proliferation induced by LPS in vivo, it suggested that LPS might have collapsed the cell cycle of the endometrium in vivo and in vitro. Thus, in the present study, we showed that LPS inflammation disrupts the cell cycle and alters cell proliferation. The proliferation marker Ki67 is recognized as a diagnostic marker for uterine smooth muscle tumors [41]. Therefore, the abnormal expression of Ki67 observed in our study is noteworthy not only for the diagnosis of CE but also for its potential contribution to related phenomena in gynecological cancers.

In conclusion, our study unveils that the inflammatory signaling triggered by LPS is implicated in the initiation of chronic endometritis (CE), as LPS stimulates both inflammatory responses and the cell cycle within the endometrium (Figure 7). Moreover, our identification of LPS and IL-6 in CE underscores their potential as appropriate diagnostic criteria. While a plausible correlation between LPS and CE exists, establishing a direct causal relationship necessitates further thorough investigation and substantiation. These findings not only contribute significantly to unraveling the mechanism of CE but also offer valuable insights into the potential treatment of gynecological diseases. In the future, integrating these markers with hysteroscopy and histological examination holds promise for enhancing the precision of CE diagnoses in clinical practice.

## 4. Materials and Methods

### 4.1. Cell Culture

EM-E6/E7/hTERT-2, a human uterine endometrial cell line (JCRB1545, Nibiohn, Japan), was cultured in Dulbecco’s modified Eagle’s medium (DMEM)/Ham’s F-12 (048-29785, Fujifilm Wako, Tokyo, Japan) supplemented with 10% fetal bovine serum (FBS) (OAC-001, Lot:171220, Japan Bio Serum, Fukuyama, Japan) and 1% penicillin–streptomycin (168-23191, Fujifilm Wako, Japan), at 37 °C in a 5% carbon dioxide-containing atmosphere. The cells were inoculated in 6-well plates (TR5000, Nippon Genetics, Tokyo, Japan) at a density of 5.5 × 10^5^ cells/well and allowed to become confluent. After serum starvation, LPS (serotype O111:B4, Sigma-Aldrich, St. Louis, MO, USA) and/or E2 10^−10^ mol/mL (E2758, Sigma-Aldrich, USA) were added to the culture medium. Four hours after treatment, the cells were collected in a microtube using TRIzol reagent.

### 4.2. Cell Proliferation Assay

Cell proliferation was assessed using a cell counting kit-8 (CCK8) assay (343-07623, Dojindo, Kumamoto, Japan) according to the manufacturer’s protocol. The cells were plated at a density of 5.0 × 10^3^ cells per well in 96-well flat-bottom plates (167008, Thermo Fisher Scientific, Waltham, MA, USA). Cells were cultured in the absence or presence of LPS and/or E2 10^−10^ mol/mL without FBS for 4 h. After treatment, 10 μL of CCK8 assay solution was added to each well and incubated for another 4 h. The signal was recorded on a microplate spectrophotometer (Multiskan GO, Thermo Fisher Scientific, USA) to measure absorbance at 450 nm wavelength.

### 4.3. Animal Treatment

Ten female ICR mice (Jackson Laboratory, Tsukuba, Japan) were used in this study. The mice were 6–7 weeks old, 30 ± 5 g in weight, fed a standard diet, and housed in a temperature-controlled room (23 ± 2 °C) and humid (50 ± 5%) environment with 12 h light/12 h dark cycle. This study was performed under the Regulations Regarding Animal Experiments of the Obihiro University of Agriculture and Veterinary Medicine (approve number 22-173). The mice were randomly divided into control (n = 5) and LPS (n = 5) groups. The body weight, food intake, and water consumption of the mice were measured once weekly. The method of inducing endometritis in mice was similar to that used in previous studies [12,42]. Briefly, 50 μL of LPS (1 mg/mL) was injected in the vagina of the mice in the LPS group once a week for 2 weeks. After a week, the mice were anesthetized using isoflurane, blood from their heart was collected for measurement of LPS concentration, and uterus samples were used for evaluating gene expression and histological analysis. Collected samples were stored at –80 °C until experiments were performed.

### 4.4. Real-Time PCR

RNA isolated from endometrial tissues and cells was treated with DNase I (18068-015, Thermo Fisher Scientific, USA). cDNA was reverse-transcribed using SuperScript II (18064022, Thermo Fisher Scientific, USA) following the manufacturer’s instructions. The primer sequences used are listed in Table 1. Real-time PCR was performed using SsoAdvanced Universal SYBR Green Supermix (1725271, Bio-Rad, Hercules, CA, USA) on a LightCycler 96 system (05815916001, Roche, Rotkreuz, Switzerland). The temperature condition for PCR amplification was followed by 35 cycles consisting of 95 °C for 10 s, 60 °C for 60 s. *β-actin* (*ACTB*) was used as a reference gene, and the relative gene expression was analyzed using the 2^−ΔΔCt^ method.

### 4.5. Histological Analysis

Histological samples were collected from mouse uterus tissue. The uterine samples were fixed in 10% formalin, dehydrated using ethyl alcohol series, cleared in xylene, and embedded in paraffin wax. Paraffin-embedded samples were sectioned to a thickness of 4 μm using a SM2000R microtome (Leica, Wetzlar, Germany). The sections were air-dried, deparaffinized, and stained with hematoxylin and eosin. Images were obtained using a ZEISS Axio Zoom (ZEISS, Jena, Germany). V16 for Bioligy microscope (Carl Zeiss AG, Oberkochen, Germany) with ZEN 3.1 Pro software (Carl Zeiss AG, Germany). The length of the uterine cavity, dense layer, cavernous layer, and myometrium were measured using ImageJ Fiji (https://imagej.net/software/fiji/ (accessed on 29 December 2023)). The measurements were repeated on 10 different uterine sections for each sample, and the measurements were averaged.

### 4.6. Ethical Considerations

This study was conducted in accordance with the tenets of Declaration of Helsinki after obtaining permission from the Obihiro University of Agriculture and Veterinary Medicine (Ethics review document: 2020-05-2). Informed consent was obtained from all the patients prior to the collection of tissue samples.

### 4.7. Study Participants and Design

Endometrial curettage tissues and blood samples were collected from patients with suspected CE (n = 13) and controls who were not suspected of CE (n = 15) from the Obihiro ART Clinic (Hokkaido, Japan). Patients with suspected CE were defined as those with endometrial polyps detected on transvaginal ultrasonography performed by a single physician. The clinical criteria were superficial stromal edema, increased stromal density, and pleomorphic stromal inflammatory infiltrate dominated by plasma cells.

Endometrial tissues were collected from patients aged 27–42 years by dilation and curettage between 2020 and 2022. Tissue was removed from the uterine cavity by curettage or aspiration biopsy. Patient data and CD138 immunostaining results were obtained from the ART clinic. Endometrial biopsy specimens were obtained randomly from patients, and histological examination of CD138 was assessed by a single pathologist who was blinded to the CE findings. Immunohistochemical (IHC) staining was performed using the CD138 antibody (M7228, Dako, Santa Clara, CA, USA), and DAB staining was performed using DAB (109431, Roche, Switzerland) according to the manufacturer’s instructions.

Samples were collected in Protein LoBind Tubes (EP0030108116, Eppendorf, Hamburg, Germany). Collected endometrial tissue was added to 300 μL of phosphate-buffered saline (PBS), vortexed, centrifuged at 3000 rpm for 10 min at 4 °C, and separated into supernatant fluid and tissue. Endometrial tissue was homogenized and stored in TRIzol reagent (15596018, Thermo Fisher Scientific, USA) at −80 °C until RNA extraction was performed. Blood was collected in a heparin-added vacuum blood collection tube, transferred to a 1.5 mL tube, and centrifuged at 3000 rpm for 10 min at 4 °C. Plasma was collected and stored at −80 °C.

### 4.8. LPS Measurement Using Limulus Amebocyte Lysate (LAL) Assay

Plasma and supernatant fluid from endometrial tissue were centrifuged at 3000 rpm for 20 min at 4 °C. These samples were diluted 10-fold with 0.02% TritonX-100 (162-24755, Fujifilm Wako, Japan) and endotoxin-free water and then heat-inactivated at 70 °C for 10 min.

Endotoxin concentrations were measured using PYROSTER NEO (294-36731, Fujifilm Wako, Japan). First, 25 µL of pretreated sample and 25 µL of endotoxin detection reagent were added to each well of a 96-well flat-bottom plate (44-2404-21, Thermo Fisher Scientific, USA). The plate was read using a microplate reader (SH-1000, Corona Electric, Hitachinaka, Japan). The measurement conditions of the microplate reader were set as follows: temperature: 37 °C, measurement mode: kinetic, onset OD: 0.015, auto mix: once, wavelength: 405 nm (main wavelength). A standard control endotoxin, *E. coli* UKT-B strain reagent (293-16541, Fujifilm Wako, Japan), was used for the standard curve.

### 4.9. Statistical Analysis

All data were presented as the mean ± standard error of the mean (SEM). Statistical analysis was performed using free software R Version 4.1.3 (https://www.r-project.org/ (accessed on 29 December 2023)) or Microsoft Excel (https://www.microsoft.com/en-sg/microsoft-365/excel (accessed on 29 December 2023)). The results of each experiment were compared with those of the control using Mann–Whitney’s U-test or t-test. Dunnett’s test was used to compare the means of several experimental groups with the mean of the control group in the experiments with EM cells. In the experiments using human tissue, a correlation coefficient was calculated using Spearman’s rank correlation coefficient (rs). Patient data were expressed as percentages and analyzed using Fisher’s exact test. Statistical significance was set at *p* < 0.05.

## 5. Patents

This research report has been issued the patent number 2023-058245 in Japan.

## Figures and Tables

**Figure 1 ijms-25-02017-f001:**
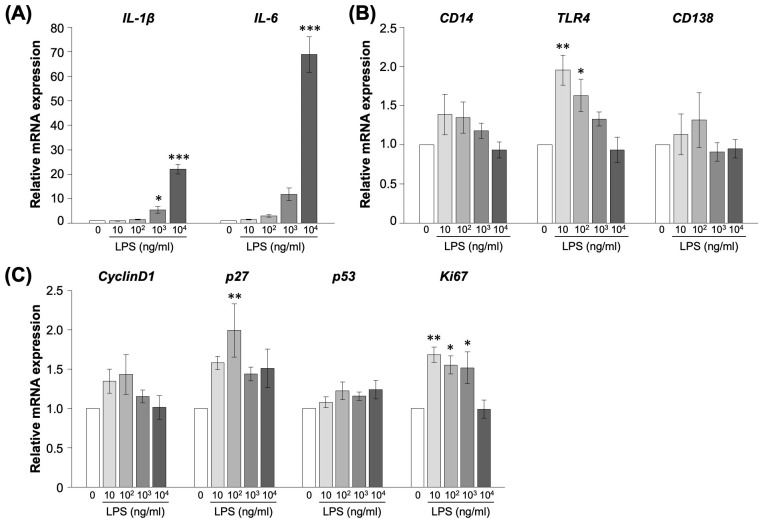
Gene expression of inflammatory and cells cycle markers in EM-E6/E7/hTERT-2 cells. The cells were administrated LPS (0, 10, 10^2^, 10^3^ and 10^4^ ng/mL) for 4 h. The experiments were conducted more than twice. (**A**,**B**) Inflammatory markers: *IL-1β*, *IL-6*, *CD14*, *TLR4* and *CD138*. (**C**) Proliferation markers: *CyclinD1*, *p27*, *p53* and *Ki67*. Values are showed as mean ± SEM, * *p* < 0.05, ** *p* < 0.01, *** *p* < 0.005.

**Figure 2 ijms-25-02017-f002:**
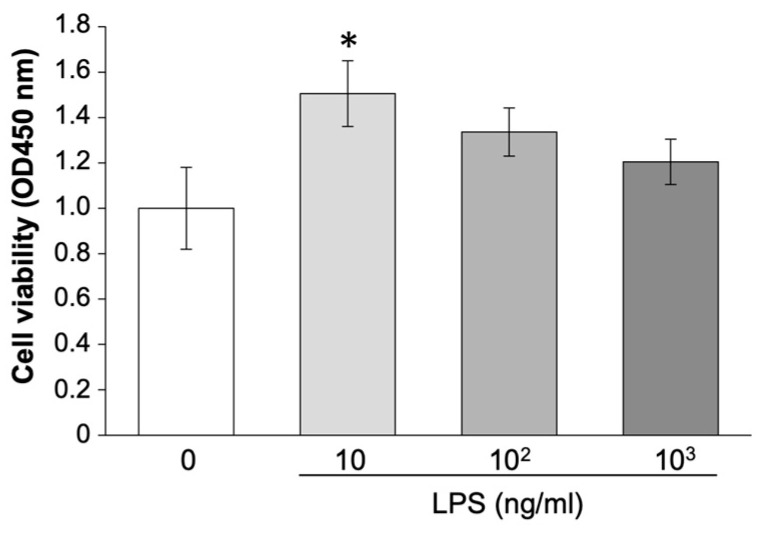
CCK8 assay for proliferation analysis using EM-E6/E7/hTERT-2 cells. The cells were administrated LPS (0, 10, 10^2^ and 10^3^ ng/mL) for 4 h. Values are shown as mean ± SEM (n = 5) and * *p* < 0.05.

**Figure 3 ijms-25-02017-f003:**
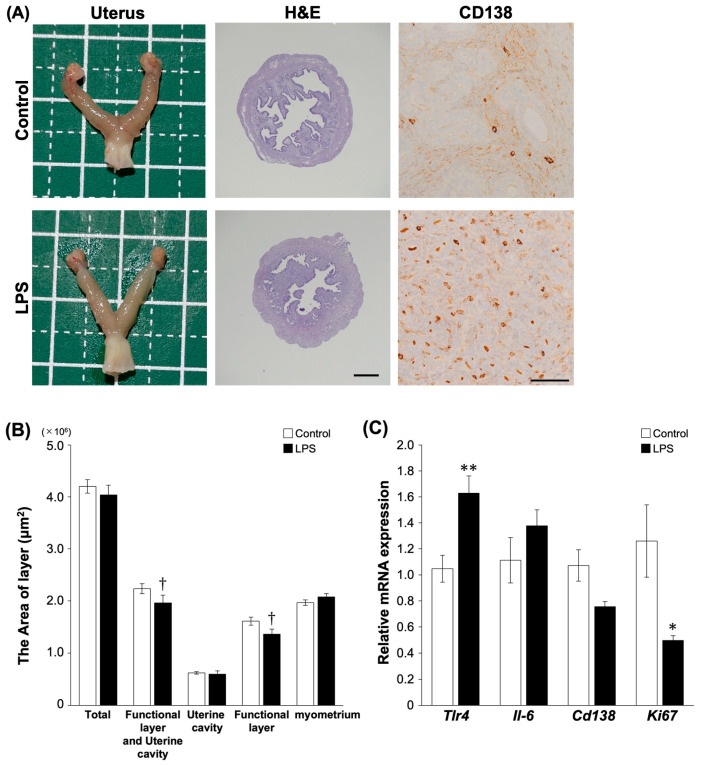
Effects of consecutive LPS administration on mouse uterus in vivo. (**A**) Histology of uterus after LPS administration. Magnification H&E staining (×40) with Scale bar = 50 μm and CD138 (×40) with Scale bar = 10 µm. (**B**) Area of total, functional layer and uterine cavity, uterine cavity, functional layer and myometrium compared between control (n = 5) and LPS group (n = 5). Values are showed as mean ± SEM. (**C**) Gene expression related with CE between control and LPS group in mouse uterus. The experiments were conducted twice. Values are showed as mean ± SEM and † *p* < 0.1, * *p* < 0.05, ** *p* < 0.01.

**Figure 4 ijms-25-02017-f004:**
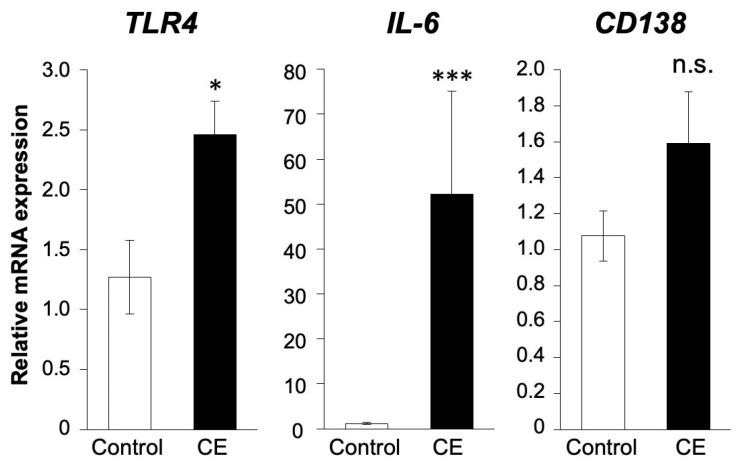
Gene expression of inflammatory markers in human endometrial tissue. Comparison of expression of *TLR4*, *IL-6*, and *CD138* genes between controls (n = 10) and CE patients (n = 13). The experiments were conducted once, utilizing uterine endometrial tissue samples from each patient. Values are showed as mean ± SEM, n.s.: not significant, * *p* < 0.05, *** *p* < 0.005.

**Figure 5 ijms-25-02017-f005:**
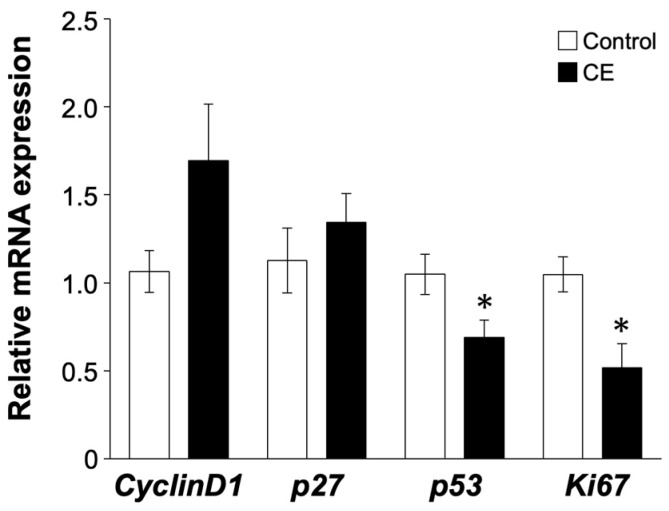
Gene expression of cell cycle markers (*CyclinD1*, *p27*, *p53*, *Ki67*) in human endometrial tissue. Comparison of gene expression between controls (n = 10) and CE patients (n = 13). The experiments were conducted once, utilizing uterine endometrial tissue samples from each patient. Values are shown as mean ± SEM and * *p* < 0.05.

**Figure 6 ijms-25-02017-f006:**
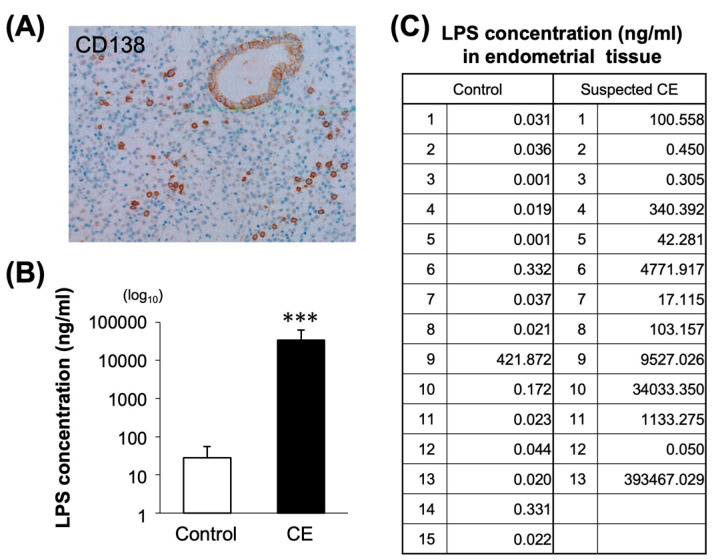
Diagnosis of CE and LPS concentration levels in human endometrial tissue supernatant. (**A**) Immunohistochemistry of CD138 in CE patient. Plasma cells stained with anti-CD138 (brown color). Magnification (×20). (**B**) LPS concentration of endometrium between controls (n = 15) and CE patients (n = 13). (**C**) Individual LPS concentration between controls and CE. Values are shown as mean ± SEM, *** *p* < 0.005.

**Figure 7 ijms-25-02017-f007:**
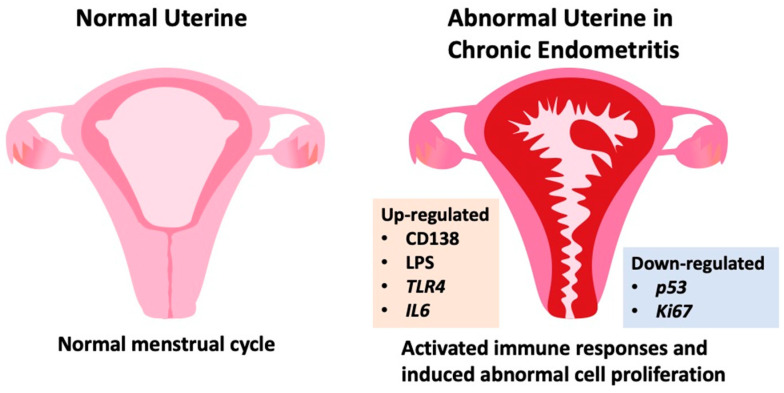
Simple graphical abstract for CE induced abnormal factors.

**Table 1 ijms-25-02017-t001:** Primer pairs used in gene expression analysis.

Gene	Primer		Size (bp)	Annealing Temperature (°C)	Accession No.
Human *TLR4*	F	TCCTGCGTGAGACCAGAAAG	125	60	NM_003266.4
	R	AATGGAATCGGGGTGAAGGG			
Human *CD14*	F	AAGCACTTCCAGAGCCTGTC	138	60	NM_000591.4
	R	TCGTCCAGCTCACAAGGTTC			
Human *CD138*	F	GGCTACTAATTTGCCCCCTGA	163	60	NM_002997.5
	R	TTCTGGAGACGTGGGAATAGC			
Human *IL-1β*	F	AACAGGCTGCTCTGGGATTC	174	60	NM_000576.2
	R	GTCCTGGAAGGAGCACTTCA			
Human *IL-6*	F	CTCACCTCTTCAGAACGAATTG	148	60	NM_000600.5
	R	CCATCTTTGGAAGGTTCAGGTTG			
Human *CyclinD1*	F	TGACCCCGCACGATTTCATT	143	60	NM_053056.3
	R	CATGGAGGGCGGATTGGAAA			
Human *p27*	F	GGCCTCAGAAGACGTCAAAC	227	60	NM_004064.5
	R	CATCCAACGCTTTTAGAGGCAG			
Human *p53*	F	GAGGTTGGCTCTGACTGTACC	133	60	NM_000546.6
	R	TCCGTCCCAGTAGATTACCAC			
Human *Ki67*	F	GTGGAAGTTCTGCCTACGGA	237	60	XM_011539818.2
	R	TAGTGCCCAATTTCTCAGGC			
Human *ACTB*	F	GGATTCCTATGTGGGCGACGA	282	60	NM_001101.5
	R	GCGTACAGGGATAGCACAGC			
Mouse *Tlr4*	F	CGCTGCCACCAGTTACAGAT	263	60	NM_021297.3
	R	CTTCAAGGGGTTGAAGCTCAGA			
Mouse *Il-6*	F	GGATACCACTCCCAACAGACC	251	60	NM_001314054.1
	R	GGTACTCCAGAAGACCAGAGGAA			
Mouse *Cd138*	F	TGACTCCAGCCGGCGAAA	299	60	NM_011519.2
	R	AAGTTGTCAGAGTCATCCCCA			
Mouse *Ki67*	F	GAGGCTGAGACATGGAGACATA	245	60	NM_001081117.2
	R	TATCTGCAGAAAGGCCCTTGG			
Mouse *Actb*	F	CGTGCGTGACATCAAAGAGAA	201	60	NM_007393.5
	R	TGGATGCCACAGGATTCCAT			

## Data Availability

Data is contained within the article and Appendix A.

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
