# Peer review of "Evaluation of Lipopolysaccharide and Interleukin-6 as Useful Screening Tool for Chronic Endometritis"

_ijms, 2024, doi:10.3390/ijms25042017_

Round 1

Reviewer 1 Report

Comments and Suggestions for Authors

Dear editors and authors:      

        It is a great honor and pleasure for me to be invited as the reviewer for this important work entitled “Evaluation of lipopolysaccharide and interleukin-6 as diagnostic biomarker for chronic endometritis”. Erina Yoneda and co-authors investigated the application of inflammatory biomarkers to diagnose chronic endometritis (CE). This study topic is interesting, attributing to their team’s long-term efforts and contributions in this scientific field. Although the experiment was well performed, I have a number of comments concerning this study:

1.     In light of the study design, the results could be interpreted as “correlations between indicated biomarkers and CE” or “patho-mechanisms”. Since the receiver operating characteristic (ROC) analysis to assess the accuracy of model predictions is lacking, the title should be rephrased. “Diagnostic biomarker” should be replaced by “useful screening tool” or “assistant tools”.

2.     The diagnosis of chronic endometritis is complicated, there are mainly three detection methods, and cross-testing will also be performed according to the clinical situation: (1). Tissue examination: After sectioning the endometrial tissue, its specific markers are detected by immunostaining. Limitations: Diagnostic results may not be accurate depending on the testing site and menstrual cycle. (2). Hysteroscopy: Observe edema and thickness through the hysteroscope, as well as whether there are polyps and other conditions

Limitations: The doctor’s subjective judgment will affect the test results.

(3). Bacterial culture: detect the cause of infection (such as GPC (Enterococcus, Streptococcus, Staphylococcus), GNB (Escherichia coli), atypical pathogens (Mycoplasma), etc.) Limitations: Some bacterial species are difficult to culture, and there are also bacteria that will affect the culture results. Most of all, LPS serves as the product of GNB alone, not GPC. The differential diagnosis is not comprehensive. The above description should be discussed in the text.

3.     The conclusion: the term of diagnostic biomarker should be rephrased as “biomarkers for differential diagnosis” or other terms.

4.     Ther cell proliferation markers are also the biomarkers for cancer. How about investigating the correlation/diagnosis/mechanism of endometrial cancer in this study design? Please discuss this issue.

5.     A simple graphical abstract should be provided.

The research is interesting that should be published soon after appropriate revision.

Comments on the Quality of English Language

Extensive editing of English language is required.

Author Response

We sincerely appreciate the invaluable comments provided by the reviewer, which greatly contributed to the significant improvement of our paper. The manuscript has been meticulously revised and reorganized based on their feedback, resulting in a more enhanced scientific framework.

  1. In light of the study design, the results could be interpreted as “correlations between indicated biomarkers and CE” or “patho-mechanisms”. Since the receiver operating characteristic (ROC) analysis to assess the accuracy of model predictions is lacking, the title should be rephrased. “Diagnostic biomarker” should be replaced by “useful screening tool” or “assistant tools”.

>> Thank you for your guidance. Taking your valuable advice into consideration, we have adopted "useful screening tool." The new title is "Evaluation of lipopolysaccharide and interleukin-6 as a useful screening tool for chronic endometritis." 

  1. The diagnosis of chronic endometritis is complicated, there are mainly three detection methods, and cross-testing will also be performed according to the clinical situation: (1). Tissue examination: After sectioning the endometrial tissue, its specific markers are detected by immunostaining. Limitations: Diagnostic results may not be accurate depending on the testing site and menstrual cycle. (2). Hysteroscopy: Observe edema and thickness through the hysteroscope, as well as whether there are polyps and other conditions

Limitations: The doctor’s subjective judgment will affect the test results.

>> As noted by the reviewer, the impact of subjective judgment by physicians on test results is a crucial point to consider. In light of this observation, we have incorporated the following statement into the background section.

p. 1-2, l. 44-51

” Hysteroscopy provides subjective information by the physician, which may not confirm the clinical findings. Pathological diagnosis of CE involves staining for plasma cells in endometrial tissue, which is frequently performed using CD138 immunostaining. However, the histological method of CD138 cannot be used in all the scenarios due to lack of consensus on a threshold for the definition of CE [8]. Additionally, the efficiency of CD138 detection depends on the timing of menstrual cycle, which influences endometrial proliferation [5,9]. Therefore, both hysteroscopy and pathological examination remain uncertain for the diagnosis of CE.”

  1. Bacterial culture: detect the cause of infection (such as GPC (Enterococcus, Streptococcus, Staphylococcus), GNB (Escherichia coli), atypical pathogens (Mycoplasma), etc.)

Limitations: Some bacterial species are difficult to culture, and there are also bacteria that will affect the culture results. Most of all, LPS serves as the product of GNB alone, not GPC. The differential diagnosis is not comprehensive. The above description should be discussed in the text.

>> Considering that the identification of certain bacteria can be detecting due to difficulties in cultivation, we have acknowledged the inability to detect some species. In recent years, Next Generation Sequencing (NGS) has gained attention as an alternative useful method, and this has been mentioned in the text as follows.

p.1, l. 42-44

In general, CE is diagnosed using hysteroscopy and pathologic examination. Currently, next generation sequencing (NGS) analysis is focused on the bacterial flora of the vagina and uterus [5,7].

p.8, l. 248-254

Some bacterial species may pose challenges in cultivation, and certain bacteria can influence the accuracy of culture results. These considerations should be thoroughly discussed in the context of chronic endometrial diseases during further exploration. Additionally, it is essential to incorporate information on NGS analysis, as it relates to the study of CE [5]. NGS provides a powerful tool for comprehensive microbial profiling and may be offer valuable insights into the microbial composition and diversity associated with CE.

  1. The conclusion: the term of diagnostic biomarker should be rephrased as “biomarkers for differential diagnosis” or other terms.

>> Thank you for your suggestion. We have rewritten the wording.

p.8, l. 246

“biomarkers for differential diagnosis”

  1. The cell proliferation markers are also the biomarkers for cancer. How about investigating the correlation/diagnosis/mechanism of endometrial cancer in this study design? Please discuss this issue.

>> We appreciate your encouraging remarks that highlight the future potential of our research. I have added the statement suggesting the potential relationship between the cell proliferation biomarker and the cancer that chronic endometritis may induce.

p.9, l. 276-280

The proliferation marker Ki67 is recognized as a diagnostic marker for uterine smooth muscle tumors [41]. Therefore, the abnormal expression of Ki67 observed in our study is noteworthy not only for the diagnosis of CE but also for its potential contribution to related phenomena in gynecological cancers.

6. A simple graphical abstract should be provided.

>> We have added a simple graphical abstract to facilitate understanding.

p.9, l. 293

“Figure7. Simple graphical abstract for CE induced abnormal factors"

Reviewer 2 Report

Comments and Suggestions for Authors

Lipopolysaccharide (LPS) is a well-known endotoxin comprising a lipid and a polysaccharide composed of O-antigen, outer core, and inner core joined by a covalent bond. LPS is a component of the cell wall of Gram-negative bacteria (e.g., Escherichia coli, Ureaplasma urealyticum, and Gardnerella vaginalis). LPS is an antigen that induces immune responses in uterine endometrial cells and can trigger an imbalance in cytokines in the uterine endometrium. This article is a valuable material that analyzes LPS effects on endometrial cells. The results obtained may sustain the link between chronic endometritis and LPS. Prior studies analyzed LPS as an inflammation promoter but also as a marker of inflammation.

I have a few minor issues to be addressed by the authors.

1.       Please see the following reference and cite it in lines 56-58.

Kim W, Choi J, Yoon H, Lee J, Jun JH. Detrimental effects of lipopolysaccharide on the attachment and outgrowth of various trophoblastic spheroids on human endometrial epithelial cells. Clin Exp Reprod Med. 2021 Jun;48(2):132-141. doi: 10.5653/cerm.2021.04448. Epub 2021 May 31. PMID: 34078006; PMCID: PMC8176151.

2.       The conclusions have a reduced foundation. The authors state that the inflammatory signaling evoked by LPS leads to the onset of CE, since LPS stimulates the inflammatory responses and cell cycle in the endometrium. These findings are important for uncovering the mechanism of CE. Additionally, these would also provide insights into the treatment of gynecological diseases.’

I believe that it is fair to declare that LPS may be correlated with CE, but it is a long way to proving that LPS leads to CE onset.

Comments on the Quality of English Language

The English language is satisfactory.

Author Response

We sincerely appreciate the invaluable comments provided by the reviewer, which greatly contributed to the significant improvement of our paper. The manuscript has been meticulously revised and reorganized based on their feedback, resulting in a more enhanced scientific framework.

  1. Please see the following reference and cite it in lines 56-58.

Kim W, Choi J, Yoon H, Lee J, Jun JH. Detrimental effects of lipopolysaccharide on the attachment and outgrowth of various trophoblastic spheroids on human endometrial epithelial cells. Clin Exp Reprod Med. 2021 Jun;48(2):132-141. doi: 10.5653/cerm.2021.04448. Epub 2021 May 31. PMID: 34078006; PMCID: PMC8176151.

>> We quoted the statement as follows and provided an explanation regarding the relationship between LPS and epithelial cells in the background.

p.2, l.57-58

In humans, LPS affects the trophoblastic spheroids and endometrial epithelial cells and decreases uterine receptivity [14].

  1. The conclusions have a reduced foundation. The authors state that ‘the inflammatory signaling evoked by LPS leads to the onset of CE, since LPS stimulates the inflammatory responses and cell cycle in the endometrium. These findings are important for uncovering the mechanism of CE. Additionally, these would also provide insights into the treatment of gynecological diseases.’

I believe that it is fair to declare that LPS may be correlated with CE, but it is a long way to proving that LPS leads to CE onset.

 >> Thank you for your valuable feedback on the conclusion of this paper. As a result of the study, we have incorporated insights into the summary, emphasizing that LPS stimulates inflammatory responses and the cell cycle within the endometrium.

p. 9, l. 281-290

In conclusion, our study unveils that the inflammatory signaling triggered by LPS is implicated in the initiation of chronic endometritis (CE), as LPS stimulates both inflammatory responses and the cell cycle within the endometrium (Figure 7). Moreover, our identification of LPS and IL-6 in CE underscores their potential as appropriate diagnostic criteria. While a plausible correlation between LPS and CE exists, establishing a direct causal relationship necessitates further thorough investigation and substantiation. These findings not only contribute significantly to unraveling the mechanism of CE but also offer valuable insights into the potential treatment of gynecological diseases. In the future, integrating these markers with hysteroscopy and histological examination holds promise for enhancing the precision of CE diagnoses in clinical practice.

Round 2

Reviewer 1 Report

Comments and Suggestions for Authors

The authors work hard to achieve the reviewers' requirements.

Comments on the Quality of English Language

Minor editing of English language is required.